# The Influence of Individual and Contextual Factors on the Vocational Choices of Adolescents and Their Impact on Well-Being

**DOI:** 10.3390/bs13030233

**Published:** 2023-03-07

**Authors:** Stefania Fantinelli, Ciro Esposito, Leonardo Carlucci, Pierpaolo Limone, Francesco Sulla

**Affiliations:** 1Department of Human Studies, University of Foggia, 71121 Foggia, Italy; 2Pegaso Online University, 80143 Napoli, Italy

**Keywords:** vocational choices, youth development, career congruence, career guidance, well-being

## Abstract

Adolescents who have to make decisions regarding their future career or academic path can be greatly influenced by parental expectations and other individual and contextual factors. The aim of this study is to explore the impact of adolescent–parent career congruence on adolescents’ well-being and future intention to enroll in a university course. The recruitment of participants took place through a combination of convenience sampling and snowball sampling. A sample of 142 high school students who are managing their decisions for the future completed an online questionnaire. Data were analyzed through a path analysis (SEM) with observed variables, and different indices were evaluated to check the model goodness of fit. The data show that congruence with parents’ wishes has a significant effect on academic motivation, work hope and mattering, which in turn have a positive and significant effect on both future intentions to undertake university studies and on the participants’ occupational well-being. In line with past studies, our results demonstrate correlations between adolescent–parent career congruence in career exploration and decision making, pointing out in particular the influence due to complementary congruence with mothers. Furthermore, our study underscores the important role played by both individual and contextual factors in adolescent well-being and intentions for their future. Finally, implications for the practice of vocational guidance practitioners are discussed.

## 1. Introduction

During adolescence there are many critical changes and challenges, such as the definition of self-identity and aspirations for the future [1,2]. Adolescents attending their last year of high school have to make decisions for their future career or academic path. Several individual and contextual factors affect youth decision making. Some of these factors, such as cognitive and psychosocial development, impact the timing and ability of decision-making competence, as well as the process by which decision making occurs. Other factors, such as parent influences, impact the process of decision making as well as specific components of the decision process [3]. These factors are interrelated. Research has shown that during adolescence, grey matter, the tissue in the frontal lobe responsible for our ability to think, is reduced or selectively ‘pruned’. At the same time, a myelination process happens, where the white matter in the brain ripens to function more effectively. As such, the aspects of the brain responsible for decision making and other aspects of psychosocial maturity are not fully developed until young adulthood, with males developing even more slowly than females [4]. For this reason, parents might be even more salient to adolescents, especially when it is time for their very first important decision, one that is linked to vocational choices: the transition from secondary school to tertiary education or the labour market.

Literature concerning vocational interests and choices often makes references to the social cognitive career theory (SCCT) [5], which represents a model for the understanding of vocational interest generation, career choices and performance, and it is the theoretical framework for the present study. The social cognitive career theory is a triadic model based on three key points: person, behavior and situation, which have been operationalised in different specific dimensions across the years. Several studies articulated the most common representation of the triadic structure, which is about self-efficacy, social support and goal setting [5], respectively for person, contextual and behavior frame. According to this theoretical model, there is a mutual and reciprocal interaction between the three main pillars, which in turn can affect career intentions or future behaviour.

As noted above, vocational choices can be certainly influenced by social context. As stated by Lent et al. in the social cognitive career theory, environment and subjective interactions with different social conditions affect career paths. In particular, parents’ impact can be relevant also for what concerns adolescents’ motivation [6] and well-being [7]. Positive outcomes in terms of academic motivation, career decision making and development, life satisfaction and well-being may greatly depend on the adolescents’ perception of their parents’ support [8]. In this respect, there can be a mismatch between parents and adolescents’ intentions and expectations regarding the career development of the latter; whether or not parents and adolescents agree about the adolescents’ career goals and aspirations is defined as adolescent–parent career congruence/incongruence [9], and describes the way parents and their children combine their perspectives on an important developmental task: recognising that each of them has goals and may or may not agree with each other.

According to the ecological systems theory [10], individuals will be more satisfied and fulfilled when their attitudes and behaviours find support and a match in the social environment. When personal and social environment attributes are compatible there is a fit or the so-called person–environment fit [11]. In the organizational psychology literature, two different dimensions of the fit are described: supplementary and complementary fit. The former describes a type of relation between individuals and work environment characterised by similarity in attitudes and values. On the other hand, complementary fit is more related to the skills; indeed, individuals’ abilities can be complemented by other employees on the team. In the same line of personal environment fit and in a systemic perspective, in the family context there can be two different dimensions of fit: supplementary and complementary. There is supplementary congruence when parents and adolescents are aligned with very similar expectations, values, attitudes and interests. Complementary congruence is represented by a supportive parental style; mothers and/or fathers gratify adolescents and fulfil their psychological needs, such as exploration, planning and goal setting. Moreover, in social cognitive career theory, there is great recognition for the importance of person–environment interaction, which is described in a very dynamic and mutual way: individuals’ behaviour is not just a function of person–environment interaction; rather, behaviour itself becomes a concurrent factor [12].

As far as our literature review is concerned, it is possible to assert that parents’ educational and relational style can impact adolescents’ academic motivation as well [13,14,15]. However, not all types of parent–child relationships are equally beneficial for youths’ motivation and goal pursuit. From a life span developmental approach, parent–child relationships should be adjusted according to the developmental needs of the adolescents and young and emerging adults [16]. For older youth, support that is autonomous such that parents assist in children’s problem-solving efforts foster motivation and goal progress more so than a directive or controlling style. Indeed, adolescent–parent career congruence occurs when the career needs of adolescents are met by supportive parents [17]. What is the relation between parents’ support and academic motivation though? Emadpoor et al. [18] found that perceived social support (which includes parental support) has an indirect effect on psychological well-being through the mediating role of academic motivation. Furthermore, Duineveld et al. [19] found that parents’ autonomy support facilitated students’ well-being across educational transitions. Seeing that there is an overlap between parents’ support and adolescent–parent career congruence, we hypothesise that the latter might also have an impact on students’ well-being through the mediation of academic motivation.

Well-being may be seen as a possible outcome of parental support, more specific of adolescent–parent career congruence, and also academic motivation. For example, within the framework of self-determination theory (SDT), there is a growing body of research, in settings ranging from elementary schools to postgraduate education, indicating that, when parents provide autonomy support, students exhibit greater engagement, performance and higher well-being [20]. SDT is a broad framework for understanding factors that facilitate or undermine intrinsic motivation, autonomous extrinsic motivation and psychological well-being [21].

In the same vein, from the positive psychology perspective, there can be another individual factor influencing academic motivation: work hope [22]. Hope has a key role in fostering motivation and influencing human behaviour [23]. It is a construct that includes three different dimensions: hypothesise goals, develop pathways and find strategies in order to achieve those goals [24]. This specific operationalisation of the construct makes clear that hope is meant as a personal resource, rather than a vague and abstract concept. Hope has been defined also as a possible personal source of resilience in remote working, functioning as a variable impacting work engagement [25]. Additionally, past studies demonstrated how work hope can positively affect career planning and choices [26,27], so we hypothesise that work hope could have a determining role both on the adolescents’ intention to undertake a university course and on their aspirational occupational well-being. For what concerns the individual perspective in social cognitive career theory, there are the goals dimensions, which can be defined as the grit to perform a particular activity, and it can theoretically frame the work hope and academic motivation as individual dimensions relevant in vocational choices.

Furthermore, from a more social and systemic point of view, when examining the relationship between parents and their children and the effects on well-being, it is also important to consider the feeling of mattering. Mattering is a strictly relational dimension and an essential psychological need or a crucial human motivation in several domains, such as education, work life and intimate relationships [28]. It is described as the feeling that one counts and to be important in the world, and at the same time it implies feeling valued by themselves and by others and adding value to one’s life and that of others [29,30].

Many studies underline the strong positive impact of this feeling on life satisfaction in many areas [31,32,33,34,35], also highlighting the link between mattering to family and self-esteem and well-being of young people [36].

For example, Elliott [37] found that adolescents who feel they matter to their families were much less likely to get drunk, use drugs and attempt suicide. Mattering is therefore a protective factor for the mental health of young people and it has an impact on their academic achievement and success [38,39].

Taking together both individual dimensions (work hope and academic motivation) and contextual factors (adolescent–parent career congruence and mattering), it is possible to set our study in the theoretical framework of social cognitive career theory (SCCT). [40]. One of the main assumptions of SCCT postulates that, when talking about the person-environment fit, it is crucial to have a double perspective: individuals are at the same time «products and producers of their environment» [40] (p. 362). Accordingly, adolescents may certainly receive social influence through the congruence/incongruence with parents or the experience of feeling important, but at the same time adolescents nurture their own attitudes and behaviours, such as the motivation or the desire to plan and achieve specific goals. Both contextual and individual dimensions can have an impact on adolescents’ current and expected well-being, echoing the outcome expectations described in SCCT as an individual component regarding expected consequences following specific behaviours [41]. In the same line, we can see the aspirational well-being as a possible future outcome of specific behaviour related to the intention to undertake a university path.

### 1.1. Italian Context

The Italian education and training system includes early childhood education and care (0–6 years old; non-compulsory), and primary, secondary, post-secondary and higher education. The first cycle of education includes primary (6–10 years old) and lower secondary education (11–13 years old). The second cycle of education starts at 14 years of age (courses last 5 years) and offers two different pathways: the upper secondary school education and the regional vocational training system. The first one offers general (lyceum), technical (technical institute), and vocational (professional institute) programmes, each featuring additional sub-tracks, or curricula, characterised by specific areas of study (e.g., humanities, math and science, etc.). This represents the very first choice that Italian students have to make in terms of their academic path. Their enrollment into tracks occurs largely non-selectively, that is, under a regime of open enrollment, from family choice during the final year of middle school. Students and their families may receive choice guidance by junior high school teachers and, in some cases, by orientation counsellors. Guidance includes explicit choice recommendations. However, such recommendations are non-binding [42]. At the end of the upper secondary school education, students who successfully pass a final exam receive a certificate that gives them access to tertiary education. The regional vocational training system, instead, offers three or four-year courses organised by accredited training agencies or by upper secondary schools. At the end of regional courses, learners receive a qualification that gives them access to second-level regional vocational courses or, under certain conditions, to courses at higher technological institutes—not to tertiary education. Education at higher level offers different pathways: universities, high-level arts, music and dance education institutes, higher schools for language mediators and higher technical institutes (source: Eurydice; Eurydice is a network whose task is to explain how education systems are organised in Europe and how they work. They publish descriptions of national education systems, comparative studies devoted to specific topics, indicators and statistics in the field of education. https://eacea.ec.europa.eu/national-policies/eurydice/content/italy_en (accessed on 3 March 2023)).

Regarding the students’ career development, school guidance is conceived as a task of Italian schools. The National Guidelines for lifelong guidance (Letter No. 4232/2014) recommend the guidance role of schools as a support, not only for the school-to-work transitions but in the challenging moments of life and as a promoter of employability and social inclusion. Despite the importance of guidance programs from the first cycle of education to the upper secondary level, the most efforts in guidance programs are from the secondary education to university and/or the labor market, and these activities are part of the plan for the educational offer (POF) of each school, as well as of the annual plan for continuing professional development of teachers. Similar to the process described above for the transition from first to second cycle of education, apart from families, guidance is often carried out by teachers—as stated in the 2014 National Guidelines for lifelong guidance. In addition to this, some schools also provide psychological services. Outside the schools, at the disposal of individuals, there are the territorial employment services that provide information on the opportunities for training and work [43].

Apart from the institutional sources and support for vocational guidance, it has to be noticed that the role of family—more specifically—that of parents, is extremely relevant as a cultural Italian trait. Indeed, a stereotypical Italian family nurtures familism as a prevalent value, which can be described as a model of supportive and caring family with a tendency to the identification of adolescents with their parents [44]. This cultural characteristic and past studies highlighting the relevance of both mother and father for adolescents [44,45] lead us to the decision of implementing the adolescent–parent congruence scale with doubled items (specific for mother and father).

### 1.2. Aim

The objective of the present study was to explore the impact of congruence between adolescents and their parents, both mothers and fathers, both on their career choice and perceived well-being on completion of secondary school.

In details, our main hypothesis was that parental congruence, with mother and father, has direct effects on some individual and contextual variables, such as work hope, academic motivation, and mattering, which in turn may play a role in determining the intention of adolescents to undertake a university career, as well as their well-being.

## 2. Materials and Methods

The study involved 147 Italian public high school students. The sample consisted of 112 females (76%) and 35 males (24%) with an average age of 17.9 (SD = 1.0). Regarding their parents’ education, most of the participants reported having a mother with a high school diploma (42%) and a father with a middle (31%) or a high school diploma (42%). For the main occupation, the most frequent are public employee (31%) or householder (35%) for the mother, and private employee (34%) or self-employed worker (31%) for the father. All information about the composition of the sample is reported in Table 1.

The recruitment of the sample took place through a two-stage process. During the “open week” initiative promoted by the University of Foggia in June 2022 for university orientation, the students who participated in the event were approached when attending seminar and laboratory activities and were asked to fill out an online web-based survey programmed via SurveyMonkey (convenience sampling). Participants, recruited on that occasion, were also asked to disseminate the access link to the questionnaire among their schoolmates and in general among their peers through mail and social media (virtual snowball sampling). All participants were asked to provide their consent to the processing of personal data for research purposes. Furthermore, in the case of minors, a specific authorisation to participate was requested from their parents or legal guardians.

The Ethics committee for the research in psychology of the Department of Humanities, Letters, Cultural Heritage and Educational Studies, approved the study (prot. n. 36714).

### Measures

Participants filled out an online self-report questionnaire, which included a socio-demographic section and the following scales:

Adolescent–Parent Career Congruence Scale (APCCS) [9]. It is a tool for measuring parental influence, and the original form of the scale is composed of 12 items to evaluate two sub-dimensions: complementary and supplementary career congruence with parents. In the present study, the items were doubled so that the participants could respond in regard to their relationship with their mother and father separately. Therefore, the version of the scale utilised includes 24 items and four sub-dimensions: Adolescent–Mother Career Congruence—Complementary Fit (MCC-C); Adolescent–Father Career Congruence—Complementary Fit (FCC-C); Adolescent–Mother Career Congruence—Supplementary Fit (MCC-S); and Adolescent–Father Career Congruence—Supplementary Fit (FCC-S).

Examples of items are “my mother shows me how to gain the information I need for my study/work interests” and “the progress I have made towards my study/work goals make my father happy”.

For each item, respondents have to express their degree of agreement using a six-point Likert scale, ranging from 1 (completely disagree) to 6 (completely agree).

As reported in the original validation article [9], the scale has high validity and good levels of reliability (alpha = 0.83 for the complementary congruence subscale and alpha = 0.80 for the supplementary congruence subscale).

In the present study, the alpha values are in line with those found previously: 0.88 and 0.84, respectively, for complementary congruence with the mother and with the father, and 0.90 and 0.83, respectively, for supplementary congruence with the mother and with the father.

Since the scale has not yet been validated in the Italian context, before using it we carried out a translation and back-translation work.

Work Hope Scale (WHS) [22]. It measures hope related to work, and includes 24 items, evaluating three sub-dimensions of vocational work hope: Agency (WH-A); Pathways (WH-P); and Goal (WH-G). Examples of items from the three sub-dimensions are, respectively, “I am capable of getting the training I need to do the job I want”; “My desire to stay in the community in which I live (or ultimately hope to live) makes it difficult for me to find work that I would enjoy”; “I doubt I will be successful at finding (or keeping) a meaningful job”. Each item consists of a statement with which to express degree of agreement, using a seven-point Likert scale, ranging from 1 (completely disagree) to 6 (completely agree).

In the original validation article [22], the authors report good validity and alpha values between 0.68 and 0.87 for scale subdimensions, which are in line with our results: 0.81 for Agency, 0.77 for Goals and 0.65 for Pathways.

Also in this case, the original scale was first of all translated and adapted, given the absence of a validation study of the tool in the Italian context.

Academic Motivation Scale (AMS) [46,47]. Through 20 items, it measures the motivation to engage in a course of study, including five sub-dimensions: Amotivation (AM-A), External Motivation (AM-E), Introjected Motivation (AM-IO), Identified Motivation (AM-ID) and Intrinsic Motivation (AM-II). The scale asks respondents to indicate how much the individual reasons listed can push them to study. Examples of items are “I go to school only because I have to” and “because I want to become an important person”. For the answers, a four-point Likert scale is used, ranging from 1 (not at all) to 4 (a lot).

Specifically, we used the Italian version of the scale [46], which has good characteristics of validity and reliability with alpha values for sub-dimensions ranging from 0.81 to 0.87.

In line with the authors, in our study the values of alpha ranged from good to high, with a minimum of 0.78 for Introjected Motivation to a maximum of 0.89 for Amotivation.

Mattering in Domains of Life Scale (MIDLS) [29,48]. It measures the feeling of mattering, that is, an individual’s perception of counting, of being important and of having value for themselves and for others, in their own life contexts. The scale includes 27 items that assess mattering in general and in four specific domains: self, interpersonal, occupational and community. Furthermore, for each domain, the tool measures two components of mattering: feeling valued and adding value. So, the present measure evaluates eight sub-dimensions of mattering: Self-Mattering—Feeling Valued (SM-FV); Interpersonal Mattering—Feeling Valued (IM-FV); Occupational Mattering—Feeling Valued (OM-FV); Community Mattering—Feeling Valued (CM-FV); Self-Mattering—Adding Value (SM-AV); Interpersonal Mattering—Adding Value (IM-AV); Occupational Mattering—Adding Value (OM-AV); and Community Mattering—Adding Value (CM-AV). Examples of items are: “This set of questions pertains to adding value to people close to you, such as relatives or friends. This means making a contribution to them such as helping or improving their lives in some way. When it comes to adding value to other people close to you, on which number do you stand now? …a year ago? …a year from now?”. For the answers, respondents use an 11-point Cantril scale, ranging from 0 (the minimum possible) to 10 (the maximum possible).

The Italian version of the scale [48] has good psychometric qualities and reports good levels of reliability for the scale sizes, ranging from 0.85 for to 0.97.

Our study also found similarly high alpha levels, ranging from 0.83 for Self Mattering—Adding Value to 0.95 for Occupational Mattering—Adding Value.

The Interpersonal, Community, Occupational, Physical, Psychological, Economic Well-being scale-short form (I COPPE) [49,50,51]. It measures well-being in a multidimensional perspective. The short version of the scale includes 14 items, which assess Interpersonal (I WB), Community (C WB), Occupational (O WB), Physical (PH WB), Psychological (PS WB), Economic (E WB) and Overall (OV WB) Well-being. In the reduced version of the scale, developed in the Italian context [49], for each dimension, the perception of well-being in the present and imagined for the future (one year from now), is assessed. An example of an item is “This set of questions pertains to relationships. The top number ten represents the best your life can be. The bottom number zero represents the worst your life can be. When it comes to relationships with important people in your life, on which number do you stand now?”. The scale uses an 11-point Cantril scale, ranging from 0 (the minimum possible) to 10 (the maximum possible).

All previous studies that applied this scale [49,50,51] report excellent characteristics of internal validity and good levels of reliability. In line with these studies, we found alpha indices for this scale ranging from 0.69 for Psychological Well-being to 0.87 for Economic Well-being.

Intention to undertake a university course of study index. The intention relating to future choice to attend university (INT-UN) was assessed by means of an ad hoc single item, i.e., “How much are you willing to continue your studies by enrolling at the University?”. Participants could respond using a six-point Likert scale, which ranged from 1 (not at all) to 6 (a lot).

Table 2 reports the values of α, as well as the descriptive statistics (mean and standard deviation) for each variable and the correlations between the variables hypothesised as predictors and those hypothesised as outcomes.

## 3. Results

The software IBM SPSS v.26 was used for the descriptive analyses. For each of the tools used, the sub-dimensions were computed through the sum of the observed variables. Assumptions of path analysis (e.g., departures from normality, outliers, missing data) were checked and addressed before conducting primary analyses. To test our hypothesis, we first looked at the Pearson (*r*) correlation indices to verify the presence of significant relationships between predictors and outcomes study variables. Subsequently, using the statistical package LISREL 8.7, we conducted a path analysis (SEM) [52] using covariance matrices and Maximum-Likelihood (ML) estimation methods. Path analysis with observed variables was used to determine the pathways by which sub-dimensions of adolescent–parent career congruence, work hope, academic motivation and mattering as independent variables influenced intention to undertake a university course of study and well-being as outcomes. Due to the complexity of the research design and the several variables involved, in line with our exploratory aim we tested a baseline model that included variables showing a significant association with the outcomes. Starting from the baseline model, based on the modification index tests, post-hoc step-by-step changes were then made to remove or add new paths between variables.

To check the model goodness of fit, various indices were evaluated: the absence of significance in the Chi-square test; the Root-Mean-Square Error of Approximation (RMSEA) and its 90% Confidence Interval (90% CI); the Standardised Root-Mean-Square Residuals (SRMR); the Non-Normed Fit Index (NNFI); and the Comparative Fit Index (CFI). Following the Schermelleh-Engel et al. [53] rules, we considered that: RMSEA and SRMR values less than or equal to 0.08 indicate an adequate fit, while for both NNFI and CFI, values equal to or greater than 0.95 indicate a good level of model fit.

Data were cleaned and screened for missing and out-of range values, and univariate/multivariate normality. All primary study variables were deemed to be normally distributed, since they did not exceed a kurtosis value > |7| and a skewness value > |2| (see Appendix A) [54]. No multivariate outliers were detected as assessed by Mahalanobis Distance test (below the cut-off χ^2^ = 33.25, *p* < 0.001). Regarding the hypothesis of the study, to verify the effect of the adolescent–parent career congruence on the intention to attend university and on well-being, we started by observing the correlation indices between the predictor variables and the outcomes (see Table 2). We then tested a baseline model, conducting a path analysis (SEM).

For the baseline model, the results showed unsatisfactory fit indices for the goodness of fit of this model to the data: χ^2^ (81) = 684.78 (*p* < 0.01), RMSEA = 0.23 with 90% CI [0.21, 0.24]; SRMR = 0.23; NNFI = 0.57, CFI = 0.71.

Given the lack of fit in the baseline model, we proceeded with post hoc changes, performed step-by-step and guided by the modification indices tests. To improve the model, each non-significant path was deleted in turn from the model, and a new model was re-estimated iteratively. Four other intermediate models were estimated, leading to an unidentified model.

Through this procedure, we arrived at a final model with excellent indices of goodness-fit: χ^2^ (8) = 8.65 (*p* > 0.05), RMSEA = 0.019 with 90% CI [0.00, 0.10]; SRMR = 0.044; NNFI = 1.00, CFI = 1.00. The path model obtained (see Figure 1) included the effects of adolescent–mother career congruence—complementary fit on work hope goal (β = 0.25), identified motivation (β = 0.28) and occupational mattering—adding value (β = 0.24). In turn, both work hope goal and identified motivation have significant positively effects on intention to undertake a university course of study (β = 0.24 and β = 0.38 respectively). In addition, identified motivation positively impacts also on work hope goal (β = 0.38) and OM-AV (β = 0.31). As for occupational well-being, it is positively influenced by identified motivation (β = 0.25), occupational mattering—adding value (β = 0.21) and intention to undertake a university course of study (β = 0.28). Finally, occupational mattering—adding value and intention to undertake a university course of study also affect economic well-being (β = 0.27 and β = 0.23, respectively).

## 4. Discussion

### 4.1. Theoretical Contribution

The results of the current investigation comply with the triadic structure of social cognitive career theory: person, behaviour and situation. Results highlight the impact of complementary adolescent–mother career congruence on work hope in the goal dimension, academic identified motivation and occupational mattering—adding value. The goal dimension of work hope and the identified academic motivation—as mediators—have a positive effect on the intention to undertake a university career. The identified academic motivation has positive impacts on work hope (goal dimension), occupational mattering—adding value and a direct impact also on occupational well-being. The occupational well-being as final outcome is also positively impacted by occupational mattering—adding value and by the personal future university enrollment intent. Moreover, there is an effect of both intention and occupational mattering—adding value on economic well-being.

The complementary adolescent–mother career congruence is a supportive style, aimed at the fulfilling of adolescents’ psychological needs; this variable resulted in serving in the role of predictor in our model and represents the core and the ancestral dimension of social support. Our findings are in line with past studies that have demonstrated correlations between adolescent–parent career congruence with several adolescent attitudes and behaviours relevant in the career exploration and decision making [17], but, to the best of our knowledge, this is the first empirical application of the adolescent–parent career congruence scale with separate items for mother and father. According to our results, the only significant effect was the one linked with complementary congruence with the mothers (and not with the fathers). It would be interesting to test further hypotheses (i.e., are more male or female adolescents expressing congruence with their mother?). However, as our sample is unbalanced, we cannot make hypotheses based on adolescents’ gender differences. Following this perspective, Sawitri [55] showed that female students experienced higher levels of supplementary congruence with parents, compared to male students.

As several studies have highlighted, parents have a relevant role in career and vocational choices, such as the definition of values and interests [56], the adolescents’ career choices [9], adolescents’ self-efficacy [17] and life satisfaction [8]. In addition, our research is shedding light also on multidimensional well-being, offering a first contribution to the understanding of the correlation between adolescent–parent career congruence and well-being.

At the second level of our path, we have three potential mediators that synthesise the individual and the social dimensions relevant for the two different outcomes: well-being and intention to undertake a university career.

The first mediator representing the individual level is work hope in the goal dimension: it is worth pointing out that our sample is not categorised as a marginalised group, which is the traditional kind of sample used for the investigation and evaluation of work hope [22]. As other studies proposed [57], work hope has been meant in this study as a construct central in vocational choices and able to foster achievement motivation [26].

Our result replicated the assumption made by Valero, Hirschi and Strauss [57] concerning the relation between work hope and motivation: hope may not be only an antecedent of motivation; rather, they can influence each other. Indeed, in our case, academic motivation impacts work hope. Furthermore, work hope could be a predictor of job performance and affective well-being in the workplace [57]; the role of work hope as mediator between the future time perspective and the career decision making among adolescents is also confirmed [27]. This past evidence contributes to the definition of work hope as a personal resource; more specifically in our model, the specific presence of work-related goals represents a positive booster for the future university intention.

According to social cognitive career theory, there is a strong interaction between goal, self-efficacy and outcome expectations confirming the theorisation made by Lent et al. regarding the reciprocal influence of person–behaviour–situation, which affect each other in a bidirectional way [12]. The simple presence of work-related goals can be relevant from a more practical point of view, so that vocational counsellors should investigate work hope as an attitude and in particular the goal dimension in order to foster adolescents’ motivation and self-efficacy; these dimensions represent the individual/person element of the triad.

The second mediator for the individual level in the relationship between the independent variables and outcomes in this study was the identified academic motivation. Ryan and Deci [21] defined motivation as intrinsic, extrinsic and amotivation. Identified academic motivation is a type of extrinsic motivation; it refers to the desire to accomplish something different from the activity itself (for example, obtaining a reward). The “identified” attribute is referred to the regulation type at the basis of motivation: the goal to accomplish is important for the person as something very personal and valuing, despite the behaviour being performed with an extrinsic aim [58].

Self-determination theory [21] defined motivation as both self-determined and shaped by the environment; in our case the role of the contextual conditions was played by the career congruence with parents, representing the situation element of the triad in social cognitive career theory. Apart from family context, some authors highlighted the link between motivation and environment with reference to learning: educators, just as vocational counsellors, should nurture and take care of the learning climate as an important element to raise motivation in students [59,60]. Finally, our finding is consistent with past research that underlined the relationship between academic motivation and well-being [61,62].

The third mediator is represented by occupational mattering, in particular the adding value dimension. This construct denotes the social level as adolescents perceive adding value to self and others in their primary occupations. It is interesting to note that academic motivation has an impact on occupational mattering; this relation can lead us to hypothesise that the more students are motivated in their academic course, the more they perceive it to be relevant in their environment. This assumption is supported by previous studies conducted by Vallerand [63]. Indeed, they stated that the quality of the motivation can produce changes in the outcomes; an identified regulation can be related to a higher passion toward the specific activity [64]. Finally, this specific path from academic identified motivation to occupational mattering can be a premise for future investigations concerning the impact of motivated students in their environment. Finally, as our outcome regards the students’ future intention, the behaviour as third element of the triad in social cognitive career theory could also be investigated in future studies.

### 4.2. Practical Contribution

This study has the peculiarity to merge both individual-subjective dimensions with collective-social perspectives, with the specific focus on well-being and intention to enroll in a university course.

As far as we are concerned, we believe that the study might contribute to the advancement of knowledge in this field of study in at least two directions: it is the first implementation of the adolescent–parent career congruence scale with distinct items for mother and father; there are practical implications for what concern vocational guidance interventions. Indeed, nurturing those individual psychosocial dimensions that have been seen to foster well-being and, in this way, providing support for making informed career or academic choices could become a duty for practitioners.

With a specific reference to the personal perspective explained in social cognitive career theory (SCCT) [41], there are three main components: self-efficacy, personal goals and outcome expectations. Our findings can be partly read in the light of SCCT: the self-efficacy dimension is replaced by the academic motivation as the willingness and grit to engage in a specific behaviour; personal goals are represented by the work hope construct, in particular by the goal dimension. Outcome expectations are related to the possible outcome of specific behaviour, and they can be depicted in the multiple dimensions of aspirational wellbeing (occupational and economic well-being). According to Lent, Brown and Hackett [5], the three components are the main solid pillars needed for career development.

## 5. Conclusions

The present study demonstrated the relevance of complementary congruence between mothers and their adolescent children for the latter’s future choices, as well as for their well-being. This relationship is mediated by the academic identified motivation and the work hope goal, the intention to undertake a university course of study, and occupational mattering for economic and occupational well-being. Finally, the intention about the academic future also affects the well-being levels of adolescents.

However, these results should be considered in the light of some limitations. First, the technique used for sample retrieval is not probabilistic and this severely limits the possibility of generalizing our results. In addition, the use of modification indices and a “post hoc model modification” approach represents a data-driven modification of the original hypothesised model that may not lead to the true population model. However, this approach might offer some guidance about a more complex model structure than what theory hypothesised and should be used cautiously [65].

Second, the percentage of females in the sample is very high (76%), which does not allow us to verify any gender differences in the levels of our variables. Moreover, a limited number of sociodemographic data were investigated; this might give a narrow insight regarding the cultural background of the sample. Third, the analysis of results has been performed in the light of different theoretical references because of the nature of the constructs considered in the study; social cognitive career theory was the wider guiding framework, but the specific relation among individual and contextual dimensions needed more narrow theoretical boundaries. At the same time, this variety of constructs can either open a quite unexplored stream of literature or further develop and improve the current study through future longitudinal investigations in order to investigate the link between parents’ influence, individual motivation, future career choice and actual well-being. We also see other interesting insights: according to the most recent literature, career choice and job identity are still strongly affected by gender typicality and gender stereotypes [66,67], as well as parents’ spirituality and religious orientations [68]. So, our study can be the very first step toward a more comprehensive investigation of mother and father roles in vocational choices.

Further variables that could be interesting to consider in the future are the grandparents’ role in adolescents’ life, single-parent home life and also the type of school (public, secular private, religious private). For example, a parent’s choice to enroll their children in a secular or religious school, which could be indicative of its founding values, could offer interesting data on parent–adolescent congruence.

## Figures and Tables

**Figure 1 behavsci-13-00233-f001:**
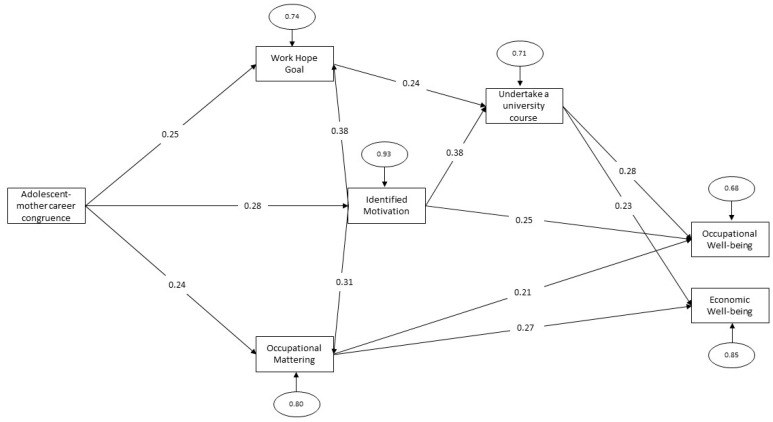
Path analysis with standardized direct effects.

**Table 1 behavsci-13-00233-t001:** Demographic characteristics of the sample (N = 147).

	M ± DS
**Age** Range	17.9 ± 1.0 16–20
	N (%)
**Sex**	
Female Male	112 (76%) 35 (24%)
**Father’s degree**	
Primary school diploma Middle School diploma High school licence Bachelor’s degree Postgraduate specialisation Missing	8 (5%) 46 (31%) 61 (42%) 23 (16%) 8 (5%) 1 (1%)
**Mother’s degree**	
Primary school diploma Middle School diploma High school licence Bachelor’s degree Postgraduate specialisation Missing	7 (5%) 33 (22%) 62 (42%) 33 (22%) 11 (8%) 1 (1%)
**Father’s employment status**	
Unemployed Public employee Private employee Self-employed worker Retired Household Missing	3 (2%) 34 (23%) 50 (34%) 46 (31%) 4 (3%) --- 10 (7%)
**Mother’s employment status**	
Unemployed Public employee Private employee Self-employed worker Retired Household Missing	5 (4%) 46 (31%) 22 (15%) 15 (10%) --- 51 (35%) 8 (5%)

**Table 2 behavsci-13-00233-t002:** Reliability indices, descriptive statistics, and correlations between hypothesised predictors and outcomes (N = 147).

Variable			INT-UN	I WB	C WB	O WB	PH WB	PS WB	E WB	OV WB
	α		--	0.81	0.86	0.83	0.84	0.69	0.87	0.76
	M ± DS		5.20 ± 1.18	7.95 ± 1.82	6.20 ± 2.30	8.03 ± 1.92	7.66 ± 2.02	6.73 ± 2.45	7.10 ± 2.09	7.54 ± 1.93
MCC-C	0.88	4.72 ± 1.53	0.23 **	0.37 **	0.27 **	0.29 **	0.31 **	0.33 **	0.17 *	0.39 **
PCC-C	0.84	3.75 ± 1.72	0.17 *	0.32 **	0.19 *	0.24 **	0.21 **	0.24 **	0.21 **	0.39 **
MCC-S	0.90	4.43 ± 1.67	0.19 *	0.30 **	0.32 **	0.26 **	0.28 **	0.31 **	0.14	0.34 **
PCC-S	0.83	3.56 ± 1.74	0.09	0.33 **	0.25 **	0.22 **	0.23 **	0.24 **	0.17 *	0.36 **
WH-A	0.81	5.06 ± 1.52	0.37 **	0.38 **	0.31 **	0.37 **	0.31 **	0.44 **	0.27 **	0.40 **
WH-P	0.65	4.74 ± 1.73	0.19 *	0.26 **	0.33 **	0.22 **	0.33 **	0.41 **	0.22 **	0.30 **
WH-G	0.77	5.01 ± 1.65	0.41 **	0.36 **	0.26 **	0.36 **	0.28 **	0.33 **	0.27 **	0.35 **
AM-A	0.89	1.43 ± 0.79	−0.20 *	−0.18 *	−0.06	−0.45 **	−0.31 **	−0.28 **	−0.17 *	−0.26 **
AM-E	0.79	2.80 ± 1.04	0.19 *	0.23 **	0.11	0.13	0.08	0.18 *	0.19 *	0.26 **
AM-IO	0.78	2.22 ± 1.10	0.11	0.07	0.10	−0.07	−0.04	0.04	−0.10	0.07
AM-ID	0.79	3.53 ± 0.71	0.49 **	0.25 **	0.20 **	0.47 **	0.24 **	0.20 **	0.20 **	0.29 **
AM-II	0.85	3.06 ± 0.86	0.40 **	0.22 **	0.15	0.38 **	0.21	0.30 **	0.12	0.31 **
SM-FV	0.83	6.75 ± 2.15	0.18 *	0.51 **	0.45 **	0.38 **	0.46 **	0.57 **	0.37 **	0.59 **
IM-FV	0.88	7.30 ± 2.28	0.15	0.51 **	0.26 **	0.35 **	0.41 **	0.49 **	0.26 **	0.54 **
OM-FV	0.85	6.79 ± 2.43	0.06	0.30 **	0.32 **	0.35 **	0.36 **	0.55 **	0.39 **	0.43 **
CM-FV	0.94	5.86 ± 2.84	0.12	0.35 **	0.53 **	0.13	0.44 **	0.52 **	0.43 **	0.46 **
SM-AV	0.89	7.59 ± 2.27	0.13	0.56 **	0.43 **	0.42 **	0.49 **	0.62 **	0.34 **	0.63 **
IM-AV	0.93	8.37 ± 1.84	0.17 *	0.45 **	0.32 **	0.29 **	0.38 **	0.41 **	0.23 **	0.36 **
OM-AV	0.95	6.69 ± 2.70	0.23 **	0.32 **	0.20 *	0.37 **	0.37 **	0.49 **	0.32 **	0.45 **
CM-AV	0.93	5.98 ± 2.65	0.10	0.35 **	0.53 **	0.20 *	0.36 **	0.43 **	0.43 **	0.39 **

Note. * *p* < 0.05; ** *p* < 0.01. α = Cronbach alpha; M = mean; SD = standard deviation; MCC-C = adolescent-mother career congruence—complementary fit; FCC-C = adolescent-father career congruence—complementary fit; MCC-S = adolescent-mother career congruence—supplementary fit; FCC-S = adolescent-father career congruence—supplementary fit (FCC-S); WH-A = work hope agency; WH-P = work hope pathways; WH-G = work hope goal; AM-A = amotivation; AM-E = external motivation; AM-IO = introjected motivation; AM-ID = identified motivation; AM-II = intrinsic motivation; SM-FV = self-mattering—feeling valued; IM-FV = interpersonal mattering—feeling valued; OM-FV = occupational mattering—feeling valued; CM-FV = community mattering—feeling valued; SM-AV = self-mattering—adding value; IM-AV = interpersonal mattering—adding value; OM-AV = occupational mattering—adding value; CM-AV = community mattering—adding value; INT-UN = intention to undertake a university course of study; I WB = interpersonal well-being; C WB = community well-being; O WB = occupational well-being; PH WB = physical well-being; PS WB = psychological well-being; E WB = economic well-being; OV WB = overall well-being.

## Data Availability

The data presented in this study are available on request from the corresponding authors. The data are not publicly available due to privacy reasons.

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
