# Peer review of "The Influence of Individual and Contextual Factors on the Vocational Choices of Adolescents and Their Impact on Well-Being"

_behavsci, 2023, doi:10.3390/bs13030233_

Round 1
Reviewer 1 Report
See the attached file.

Author Response
We wanted to thank the reviewer for their valuable comments. We now amended the manuscript accordingly, utilising the track chenages function of MS Word. Attached, you will find replies to your comments in red. Thanks for your time.

Reviewer 2 Report
Dear authors,
1. Abstract: Authors should describe data collection and analysis methods.
2. Abstract: the contents of “Strengths of this study are twofold” should be deleted and replaced with a discussion of the results of this study.
3. The discussion section is best divided into two parts. The first part is the theoretical contribution, which discusses the role of this manuscript for related research from the perspective of academic research. The second part is the practical contribution, which gives relevant suggestions to the appropriate departments and stakeholders according to the research results.
4. The manuscript lacks a conclusion section. The content of the conclusion section should include a one-by-one response to the research objectives, as well as limitations and prospects for the future.
Author Response
We wanted to thank the reviewer for their valuable comments. We now amended the manuscript accordingly, utilising the track changes function of MS Word. Attached, you will find replies to your comments in red. Thanks for your time.

Reviewer 3 Report
Comments to the authors
It is an original manuscript with an excellent research proposal. However, it has issues in its construction that need to be solved, as noted above. I encourage authors to engage in improving the manuscript, which presents a potential contribution.
My first concern relates to the introduction section.
The authors presented literature support for the research question, summarizing how some of the main theoretical approaches in career guidance have been researched and discussing the theme studied. However, they did not define a theoretical approach for the analysis of the research question, nor did they present the main concepts that would help in the discussion of the research findings. Thus, it is necessary to choose the theoretical constructs used and then conceptually define all in the manuscript from a chosen theoretical framework.
b) Second, expand the overview of the relevancy of the concepts studied in the specific cultural context of the authors.
c) And finally, provide a more detailed description of Italian culture, especially regarding education and working aspects.
My second concern relates to the method section.
a) Briefly describe how to sample specificities (variables) were controlled and discussed in the results.
b) Lack of a brief description of the context studied.
c) Methodological issue: it is necessary to present data indicating the epistemological coherence of the conceptual bases of the instruments used in the research.
d) The description of the method of data analysis should be in the method section and not in the results.
The authors presented the report of the results in a reasonable manner, but they could be described less synthetically and more broadly.
And my third concern relates to the discussion section.
a) I appreciated the discussion wherein the authors sought to explore some meaning of their results. Thus, the study is more descriptive than explanatory. Include some discussions and analytical hypotheses to make the manuscript more explanatory, improving its quality and relevance.
b) The authors should have chosen only one theoretical approach for data analysis and not used two or three theoretical references and analyzed isolated results. This procedure reduces the validity of the findings and does not allow for many hypotheses, inferences, and general correlations, as required in the discussion section.
c) It is not the authors who should keep highlighting the originality and quality of the study carried out. The relevance of the article is attributed to the scientific community through the reading and citation of the work.
Include a conclusion section, present the main original and relevant research findings, and highlight what evidence indicates this.
The aims of this paper are worthy. What is needed is a fuller realization of those aims in a more detailed, seamless, and robust report.
Author Response

(The authors gave the same response as above.)

Reviewer 4 Report
Thank you for giving me the opportunity to read this manuscript which is generally interesting and has relevant practical implications. However, some revisions may make it more suitable for publication. I try to list them following the order of the manuscript.
Introduction
1. The introduction is full of theoretical references that try to explain the choice of constructs as they arise. This is worthwhile, but leaves the reader wondering about the frame of reference within which the authors have chosen to place themselves. This framework is presented only at the end, the social cognitive career theory (SCCT). My advice is to minimally restructure the introduction, immediately clarifying to the reader that the SCCT is the reference frame and what it implies, in order to subsequently specify what the gaps are in the literature and to introduce the concepts in the light of all this.
Method and participants
2. There is an excessive gender imbalance which seems to affect the validity of the results. The authors should at least attempt to justify why they found it appropriate to base the manuscript results on this specific sample of 147 students, rather than trying to balance the gender difference.
3. There is a complete lack of information about the validity and reliability of the scales in the context of the previous literature. Also, it would be preferable to report the results of the calculation of Cronbach's alpha in this section.
4. I would invite the authors to write in the text that there is a version adapted to the Italian context of the chosen measures, to which they have referred.
Results
5. Both the analysis of the normality of the variables and that relating to the search for any outliers are absent.
6. I'm not at all sure that the authors' way of testing the path model is acceptable. Each SEM model seeks to test a theoretical model based on specific hypotheses, by comparing it to the "reality" of the data. Consequently, the initial model should not be tested on the basis of the bivariate correlations reported in Table 2, but should test the initially proposed theoretical model. Similarly, it doesn't make much sense to proceed step by step with post hoc changes based on modification rates. I understand the meaning of what has been done, but the frame of reference for these choices should always be theoretical. Furthermore, the various "nested" models should be compared to each other to understand whether or not there is a significant difference. As advice, the authors should start from a very specific theoretical and hypothetical model of relationships between the variables, then trying to arrive at a more parsimonious model by setting some relationships to zero when there is also a theoretical reason and comparing the two models gradually obtained. If the assumptions aren't so clear, then perhaps it's best to steer towards a simple exploratory regression approach.
7. The use of acronyms makes it extremely difficult to follow the flow of the text. I recommend avoiding them, possibly also in Figure 1.
Discussion
8. Most of the discussion is acceptable, but I would have expected a discussion more based on the SCCT perspective in order to present a more coherent and harmonious argument. If possible, I advise authors to go this route.
9. Finally, I suggest inserting a clear and distinct section on the limitations of the paper.
10. I am not a native Englishman, however I have the perception that the English needs to be significantly improved in some parts.
Author Response

(The authors gave the same response as above.)

Round 2
Reviewer 2 Report
Thank you for your revision.
Author Response
Thank you for your valuable comments.
Reviewer 3 Report
The authors did a great job reviewing the manuscript. However, two issues remained:
a) In the Introduction, provide a brief but detailed presentation of the triadic structure of the Social Cognitive Career Theory (SCCT): person, behavior, and situation, which were taken as the main concepts that helped the discussion of the research findings.
b) In the method, present a brief description of the context studied (high schools probably in Foggia, Italy) and the main characteristics of the students, their families, and sociocultural background from this region. That is necessary to contextualize the research findings.
Reviewer 4 Report
Thank you for this revised version of the manuscript, which seems to me improved. I would have preferred more detail on the skewness and kurtosis values of each variable. In fact, with values of these indices significantly greater than 1 in at least one variable, the literature suggests using robust estimators. Therefore, I personally consider the paper in this form acceptable, but it would be better if the authors clarify this point and, in case, use the robust estimator, as suggested.
Author Response
Please, see the attachment.
